# Foetal Growth Restriction Effects on Grey and White Matter in the Prefrontal Cortex and Basal Ganglia of Postnatal Day 10 Piglets

**DOI:** 10.3390/cells14221776

**Published:** 2025-11-12

**Authors:** Bhuvaneswari Harishankar, Kirat K. Chand, Paul B. Colditz, Julie A. Wixey

**Affiliations:** 1UQ Centre for Clinical Research, Faculty of Health, Medicine and Behavioural Sciences, The University of Queensland, Brisbane, QLD 4029, Australia; b.harishankar@student.uq.edu.au (B.H.); k.chand@uq.edu.au (K.K.C.); p.colditz@uq.edu.au (P.B.C.); 2Perinatal Research Centre, Royal Brisbane and Women’s Hospital, Herston, Brisbane, QLD 4029, Australia

**Keywords:** foetal growth restriction, brain, newborn, neuropathology, glia

## Abstract

Foetal growth restriction (FGR) is commonly caused by placental insufficiency and increases the risk of perinatal morbidity and mortality. The developing brain is vulnerable to FGR, which can result in adverse long-term neurodevelopmental outcomes. Newborn pigs with spontaneous FGR (<10th centile body weight) and normally grown (NG) littermates were reared to postnatal day 10 (P10; n = 8 FGR and n = 9 NG). Neuropathology was assessed in the prefrontal cortex (PFC) and basal ganglia (BG), which play a key role in cognitive and motor functions. FGR piglets show decreased neuronal count (NeuN) and structural integrity (MAP2) associated with increased apoptotic activity (Casp-3 and -9) in the PFC and BG. Hypomyelination was consistently observed in the white matter of the FGR brain. There was increased microglial activation (Iba-1) and GFAP-positive astrocytes in both the grey and white matter of the PFC and BG, along with increased apoptotic activity in the FGR brain. These findings suggest that the FGR piglet brain shows impaired grey and white matter associated with increased apoptosis in the PFC and BG that persists at P10. Increased glial activation and apoptotic astrocytes may impact neuronal survival and potentially contribute to adverse long-term neurodevelopmental outcomes, highlighting the need for targeted therapeutic interventions to promote effective brain repair in infants with FGR.

## 1. Introduction

Foetal growth restriction (FGR) is a significant complication in pregnancy, which increases the risk of perinatal morbidity and mortality [1,2,3]. This condition affects approximately 5–10% of all pregnancies, with an estimated 32.4 million newborns experiencing it per annum in low- and middle-income countries [4,5,6]. FGR is commonly caused by placental insufficiency, resulting in an inadequate supply of oxygen and nutrients to the developing foetus, impacting the growth and brain development of the foetus [7,8]. The brain is vulnerable to FGR, exhibiting long-term neurodevelopmental issues ranging from cerebral palsy to mild attention, learning, behavioural, and motor difficulties [9,10,11,12]. Neurodevelopmental delays are observed in up to 50% of infants with FGR at 2 years of age [13]. Studies have reported that children born with FGR exhibit impaired motor and cognitive skills at school age, which significantly affects their academic achievements [9,14,15].

Clinical imaging studies have demonstrated reduced grey matter volumes and structural alterations in white matter across multiple brain regions in infants with FGR at 12 months’ corrected age [16,17,18,19]; however, it remains unclear whether the effects of FGR are truly global or whether certain brain regions are particularly vulnerable. It is crucial to examine the changes at the cellular level in multiple regions within the same cohort to map both global and local brain structural changes in FGR. Several FGR large-animal model studies have demonstrated structural changes in the grey and white matter, including the cerebellum, hippocampus, and parietal cortex, at various time points [20,21,22,23].

The frontal cortex and the basal ganglia are brain regions responsible for motor, cognitive, and behavioural functions [24,25,26]. Infants with FGR exhibit altered network connectivity in the cortico-striatal-thalamic network, which is associated with poorer motor, cognitive, and socioeconomic performance at school age, demonstrating that these brain regions are important for neurodevelopment in infants with FGR [27,28]. These are the pivotal brain regions that begin their development prenatally through neuronal migration, differentiation, and maturation, and continue to mature postnatally through synaptogenesis and myelination, making the frontal cortex and basal ganglia vulnerable to FGR [29,30]. These regions have been studied in only a limited number of FGR animal models. Given the key roles of the prefrontal cortex and basal ganglia in the neuropathology of FGR, targeted investigation of these regions is warranted.

Previous studies, in a large animal model of FGR, have shown early cellular disruption, with neuronal and white matter injury evident from 104 days gestation (115 days term) in the piglet up to postnatal day 7 in the parietal cortex and hippocampus [21,31,32]. This study used spontaneously growth-restricted piglets as a translational model of human FGR to investigate the neuropathological changes in two relatively uncharacterised brain regions, the prefrontal cortex (PFC) and basal ganglia (BG), in the developing grey and white matter. We hypothesised that glial activation would be persistent in the PFC and BG at P10, contributing to ongoing neuronal and white matter impairment.

## 2. Methodology

### 2.1. Ethical Approval

Approval for this study was granted by the University of Queensland Animal Ethics Committee (2022/AE000519) and was carried out using National Health and Medical Research Council Guidelines and ARRIVE Guidelines for the care and use of animals for scientific purposes.

### 2.2. Animals

The piglet model is a well-established FGR model resulting from placental insufficiency, which is extensively characterised by our group and others [21,31,33]. Large white piglets were obtained on the first day of life (P1 < 18 h) from a commercial piggery. These spontaneously birthed piglets were sourced from multiple sows (nine litters) and weighed. FGR piglets (n = 8; 5 males/3 females) were identified by a birth weight below the 10th centile (mean birthweight: 0.83 ± 0.04 kg), while normally grown (NG; 40–90th centile, mean birthweight: 1.60 ± 0.05 kg) piglets (n = 9; 5 males/4 females) served as controls. Piglets were fed, cared for, and monitored at the Herston Medical Research Centre (HMRC) until P10. Piglets were euthanised by an intraperitoneal injection of sodium phenobarbital (650 mg/kg; Lethabarb, Virbac, Australia) on P10. Animals were transcardially perfused with phosphate-buffered saline (PBS), after which brain tissues were collected, weighed, and hemisected. Coronal brain slices from the right hemisphere were immersed and fixed in 4% paraformaldehyde as previously described [31].

### 2.3. Immunohistochemistry

Brain tissue samples from the right hemisphere, containing the prefrontal cortex and basal ganglia, were paraffin-embedded and sectioned at 6 µm thickness for immunohistochemical analysis (PFC—A35.0, BG—A17.5 mm, where A 0.00 represents the posterior commissure) [34]. Tissue sections were mounted on Mezel Superfrost Plus adhesive slides and dried overnight at 37 °C. All sections were subsequently dewaxed and rehydrated using standard protocols, followed by heat-induced epitope retrieval with TRIS-EDTA (pH 9) at 95 °C for 15 min and then cooled to room temperature (RT). Sections were incubated in donkey serum (5%) diluted in PBS, containing 0.5% Triton-X 100 to block non-specific binding for 1 h at RT, and primary antibodies were incubated overnight at 4 °C. Neurons were analysed using the primary antibodies, neuronal nuclei marker (NeuN, 1:1000, Abcam, Cambridge, UK (ab177487)), and microtubule-associated protein-2 (MAP2, 1:1000, Sigma-Aldrich, St. Louis, MO, USA (M4403)). Apoptotic cells were examined with caspase-9 (Casp-9, 1:1000, Abcam (ab32539)) and cleaved caspase-3 (Casp-3, 1:500, Cell signalling Technology, Danvers, MA, USA (#9661)). White matter was assessed using myelin-binding protein (MBP, 1:1000, Abcam-ab7349), pan-oligodendrocyte marker 2 (Olig2, 1:1000, Genentex, Irvine, CA, USA (GTX132732)), and neurofilament (NF, 1:1000, Abcam (ab134306)). Microglia (Ionised calcium-binding adaptor molecule-1 (Iba-1), 1:1000, Abcam (ab5076)), and astrocytes (GFAP, 1:1000, Cell Signalling (#3670)) were used as glial markers.

The incubated slides were washed with tris-buffered saline and, subsequently, incubated with species-specific secondary fluorophores along with 4′,6-diamidino-2-phenylinindole (DAPI, 1:2000) at RT for 90 min. These sections were then mounted with Prolong Gold antifade (Molecular Probes, Invitrogen Australia, Melbourne, Australia). Immunolabelling was performed in duplicate for each brain region per animal. Negative control sections, processed in parallel without primary antibodies, were included to assess non-specific binding.

### 2.4. Image Acquisition and Analysis

All tissue sections were imaged using a Zeiss Axio scope equipped with an Axiocam 503 camera (Carl Zeiss MicroImaging, GmbH, Oberkochen, Germany). Images were captured by Zen 2012 software, using an EC Plan-Neofluar 20×/0.50 M27 (FWD = 2.0 mm) objective (Carl Zeiss MicroImaging, GmbH, Germany). Grey matter of the PFC and BG, specifically the caudate nucleus (CN), putamen (P), and globus pallidus (GP), was examined. White matter was examined within the PFC (intragyral white matter), and in the BG, the internal capsule (IC), and the median medullary lamina (MML). Four fields were acquired for the PFC, and two fields of each sub-region of the BG, for analysis. Analyses were conducted on the right hemisphere, with a minimum of two replicates. Technical replicates were separated by at least 70 µm. Microglia were manually counted and classified as either ramified or amoeboid based on morphological characteristics [21,35]. Positive area labelling for astrocytes, MBP, and NF was quantified using the threshold function and moments plugin in FIJI (ImageJ v2.16.0; Image Processing and Analysis in Java; National Institutes of Health, Bethesda, MD, USA). All analysis was conducted under blinded conditions.

### 2.5. Statistics

All statistical analyses were performed using GraphPad Prism 10.0 software (GraphPad Software, San Diego, CA, USA). Data were analysed using a two-way ANOVA with Tukey’s post hoc test to compare the FGR and NG groups. Results are presented as mean ± SEM, with statistical significance accepted at *p* < 0.05. The mean inflammatory (activated microglia) score in the FGR piglet brain is 90.3 with an SD of 16.75. To detect a 30% difference between groups in inflammatory score, 8 animals per group were required (power 80%, *p* = 0.05).

## 3. Results

At P10, mean body weight was significantly lower in FGR piglets compared with NG piglets (*p* = 0.0002) (Table 1). Brain weight was also significantly reduced in FGR piglets compared with NG (*p* = 0.0003). Brain-to-body weight ratio was significantly increased at P10 in FGR piglets in comparison to NG piglets (*p* = 0.0002), indicating asymmetric growth restriction in the FGR piglets.

### 3.1. Neuronal and Structural Alterations in FGR Piglet Brains

In the current study, the NG brain showed a high number of mature neuronal cells labelled with NeuN-positive cells at P10 (Figure 1A), whereas the FGR brain showed regions that were sparse in neurons of both the PFC and BG. Quantification of NeuN-positive cell labelling (Figure 1C) showed a significant decrease in NeuN-positive cells in the FGR group compared with the NG in the PFC (*p* < 0.0001) and BG (CN: *p* = 0.0004; P: *p* = 0.0004; GP: *p* < 0.0001). MAP2 is a key component of the neuronal cytoskeleton, particularly in dendrites and perikarya (cell bodies) associated with the functions of the neurons. The NG brain displayed well-structured cell bodies with robust dendrites and axonal growth (Figure 1B) in the PFC and BG. The FGR brain showed weaker perikaryal cell body labelling with impaired dendrites and axonal growth (MAP2-positive cells) when compared to the NG brain; MAP2 labelling was measured using the area coverage that revealed significantly decreased MAP2-positive cells compared to the NG group (Figure 1D; PFC: *p* < 0.0001; CN: *p* = 0.0062; P: *p* = 0.0097; GP: *p* = 0.0295).

Changes in neuronal populations may be associated with the apoptotic activity in the newborn FGR pig (Wixey, 2019) [21]. Apoptotic activity plays a crucial role in both brain development and brain injury [36]. We first examined the initiator caspase-9 (Casp-9), which triggers intrinsic apoptotic pathways. In both the PFC and BG, we observed an increase in Casp-9-positive cells in the FGR brains (Figure 2C, PFC and BG: *p* < 0.0001). Co-localisation of NeuN-positive cells with Casp-9-positive cells demonstrated an increase in the number of neurons being signalled for cell death. In the PFC of the NG brains (Figure 2A), there is 36% co-localisation. In contrast, the FGR brain shows 60% co-localisation of NeuN-positive and Casp-9-positive cells (PFC: *p* < 0.0001). Moreover, the BG of FGR brains displayed 52–63% of co-localisation of NeuN-positive cells with Casp-9-positive cells as compared to 28–32% of NG brains. (Figure 2C, BG: *p* < 0.0001).

We next sought to determine whether this increased initiation of apoptosis resulted in more neuronal cell death. Co-localisation of cleaved caspase-3 (Casp-3)—a key downstream executor enzyme in the intrinsic apoptotic pathway—with MAP2-positive cells revealed that a higher number of MAP2-positive cells were undergoing apoptosis in FGR brains compared to NG brains (Figure 2B,D, PFC and BG: *p* < 0.0001).

### 3.2. Impaired Myelination in FGR Piglet Brains

In the present study, NG brains demonstrated well-organised, continuous myelination, associated with high numbers of Olig2-positive, as well as MBP-positive and NF-positive, labelled areas in the white matter of the PFC and BG (Figure 3A–C). FGR brains show interrupted and inconsistent MBP-positive labelling and NF-positive labelling with a lower number of Olig2-positive cells in the white matter of the PFC and BG. MBP-positive and NF-positive labelling were decreased in the white matter of the FGR brains (Figure 3D; MBP for PFC and BG: *p* < 0.0001) (Figure 3F; NF for PFC and BG: *p* < 0.0001). Further, quantification of Olig2-positive cells shows a decreased number of Olig2-positive cells/mm^2^ in the white matter of the PFC and BG in the FGR brain compared to the NG brain (Figure 3E; PFC: *p* = 0.0013; IC: *p* = 0.0153; MML: *p* = 0.0003).

### 3.3. Increased Activation of Glial Cells in the Grey Matter of FGR Brains

In the grey matter of the PFC and BG, microglial cells show dynamic morphology, which can be expressed using the microglial marker Iba-1. NG brains displayed round or oval cell bodies with symmetrical extended processes, called resting or ramified Iba-1-positive cells, distributed over the grey matter of the PFC and BG (Figure 4A). Notably, darker cell bodies—thickened and retracted processes known as activated/amoeboid Iba-1-positive cells—were more expressed in the grey matter of the FGR brains as previously reported [21,35]. Quantification of total Iba-1-positive cells was significantly increased in the grey matter of the BG of the FGR brain (Figure 4C, PFC: *p* = 0.0128, CN: *p* = 0.0024; P: *p* = 0.0264; GP: *p* = 0.0340), and activated Iba-1-positive cells were significantly increased in the grey matter of the FGR brain (Figure 4D, PFC and BG: *p* < 0.0001).

Astrocytes play a crucial role in maintaining homeostasis in the central nervous system (CNS) and promote neuronal health [37,38]. The GFAP marker was used to analyse astrocytes in the PFC and BG. GFAP-positive astrocytes show star-shaped, long, branching processes from the cell body, reflecting the normal astrocyte shape and distributed over the PFC and the BG in the NG P10 brain (Figure 4B). As previously described [21], astrocytes in the FGR brain exhibited reactive morphology, with retraction and thickening of processes into the cell body. In the FGR brain, GFAP-positive astrocyte density was significantly increased in the grey matter of the PFC and BG (*p* < 0.0001) (Figure 4E).

### 3.4. Increased Activation of Glial Cells in the White Matter of FGR Brains

Microglial cells and astrocytes were also analysed in the white matter of the PFC and BG. We observed that the total number of Iba-1-positive cells was evenly distributed across the white matter of NG brains (Figure 5A). The total number of Iba-1-positive cells was significantly increased in FGR white matter (Figure 5C, PFC: *p* = 0.0234; IC: *p* = 0.0044; MML: *p* = 0.0144). The number of activated Iba-1-positive cells was increased in the white matter of the FGR brains compared to NG brains (Figure 5D, PFC and BG: *p* < 0.0001). White matter of the FGR brains showed a higher cell density of GFAP-positive astrocytes (Figure 5B) compared to the NG brains in both the PFC and BG (Figure 5E, PFC and BG: *p* < 0.0001).

Astrocytes play a major role in neuronal health support [39]. To determine whether GFAP-positive astrocytes are also undergoing cell death in the FGR piglet, we examined the co-localisation of GFAP-positive astrocytes with Casp-3-positive cells. In the grey matter, we demonstrated elevated levels of GFAP-positive astrocytes co-localised with Casp-3-positive cells in FGR brains (Figure 6A) compared to NG brains (Figure 6C, PFC and BG: *p* < 0.0001). In the white matter, we also observed a significant increase in GFAP-positive astrocytes co-localised with Casp-3-positive cells in the FGR brains (Figure 6D, PFC and IC: *p* < 0.0001, MML: *p* = 0.0001) compared to NG brains.

### 3.5. Sex Differences

Sex differences were not observed within the phenotypes, with male and female FGR piglets displaying comparable counts in each marker assessed relative to NG piglets.

## 4. Discussion

To our knowledge, this is the first study to characterise cellular changes in the PFC and BG in a large animal model of FGR after birth at P10. This study demonstrates glial cell activation, elevated apoptosis, and neuronal and white matter alterations in P10 FGR piglet brains. Cellular pathology in the grey and white matter was consistent in both regions examined, with similar trends observed to those in our previous studies at earlier time points in the newborn FGR pig [21,31,32].

Clinical imaging studies frequently show volume reduction in the grey and white matter of infants with FGR [16,27,40]. Cellular and structural impairments at a microscopic level, as assessed by immunohistochemistry, have also been reported in large animal models of FGR [21,41,42]. Numerous studies in FGR animal models have reported a decrease in neuronal count and disrupted structural integrity at postnatal days one, four, and seven, indicating a persistent impact on neuronal development and maturation [21,31,41].

In the current study, we observed alterations to neuronal populations, as assessed by a decrease in neuronal count, altered cytoskeletal structure and structural integrity, and increased apoptosis of neuronal cells in the PFC and BG of P10 FGR piglets. The decrease in NeuN-positive cells and MAP2-positive labelling coincided with an increase in Casp-9-positive cells and Casp-3-positive cells, respectively, in FGR brains. This is similar to previous studies in the P1 and P4 FGR piglet and P1 FGR sheep [21,42]. Alterations in neuronal morphology, with increased apoptosis, were observed in late-onset FGR at P1 due to the timing and severity of hypoxic insult within the cortex [42]. These findings suggest that neuronal loss and structural degeneration are associated with activation of the apoptotic pathway. Our model is similar to late-onset FGR sheep, with neuronal changes not observed at 100/115 days of gestation, but reductions noted from 104 days of gestation [31]. Kalanjati et al. reported these findings based on MAP2 labelling of the parietal cortex and CA1 region of the hippocampus, but no examination of apoptosis was undertaken [31]. Our findings suggest grey matter may be similarly impacted irrespective of regions in the FGR pig at P10.

White matter impairment is a common neuropathological characteristic in the FGR brain. FGR sheep demonstrate impaired myelination, showing thinner myelination in the white matter at postnatal day one [43]. The total number of oligodendrocytes remains unchanged, despite the loss of mature oligodendrocytes in both early- and late-term FGR sheep, suggesting the loss of mature oligodendrocytes may plausibly contribute to decreased myelination [42]. Contrastingly, FGR guinea pigs displayed an increase in oligodendrocyte count at P1 but a reduction at 1 week of postnatal age, indicating a delayed maturation of oligodendrocytes [44]. Our data suggest persistent impaired myelination with decreased oligodendrocytes occurring in the white matter of P10 FGR, consistent with our previous studies at earlier postnatal days [21,31,35]. Oligodendrocytes play a major role in myelination by ensheathing axons and forming functional myelin, which facilitates communication between distant brain regions [45,46]. In this present study, reductions in both Olig2-positive cells and MBP-positive area coverage were observed, suggesting a loss in oligodendrocytes may contribute to hypomyelination. Olig2 is a pan-oligodendrocyte marker; therefore, oligodendrocyte lineage cannot be determined. It is plausible that predominantly immature oligodendrocytes remain but are unable to effectively myelinate these white-matter regions in FGR brains, and could potentially contribute to impaired neurodevelopmental outcomes [47]. Further analysis of oligodendrocyte lineages may help identify if specific oligodendrocyte populations are more impacted in FGR brains.

One of the key mechanisms associated with grey and white matter impairment in FGR pigs is neuroinflammation [48,49]. Neuroinflammation comprises a set of processes—including the activation of glial cells, upregulation of proinflammatory cytokines, elevated nitric oxide, and decreased anti-inflammatory cytokines—in response to the stress of injury [50,51,52]. FGR animal models have demonstrated neuroinflammation with upregulation of astrocytes and microglia at multiple time points across various brain regions, correlating with grey and white matter impairments [21,32,41,42,44,53,54].

In this study, FGR piglets exhibited an increased number of activated microglia and reactive astrocytes, which is similar to previous observations at P1 and P4 in FGR pigs [21]. Our findings also demonstrated increased co-localisation of Casp-3-positive cells and astrocytes (GFAP-positive cells) in the grey and white matter of the PFC and BG, highlighting increased astrocytic cell death at P10 in the FGR piglets. Astrocytes play a major role in mediating the inflammatory response, and also in guiding and supporting neuronal viability and synaptic transmission, which aids in brain homeostasis [39,55]. Together, astroglial reactivity and associated increased cell death may result in altered capacity to maintain the brain environment, in addition to impaired modulation of inflammatory responses, which may result in ongoing pathology in FGR brains.

Proinflammatory cytokines are the signalling molecules that play a major role in the initiation and progression of neuroinflammation. Under stress or injury, signalling molecules such as TNF-α, IL-1β, IL-6, and CCL2 are released from activated glial cells, including microglia and astrocytes. This can amplify inflammatory signals, disrupt the blood–brain barrier, and lead to grey and white matter impairment [56]. Our previous studies indicate early neuroinflammatory responses/signalling and glial impairment in the FGR brain at P1 and P4 that affect both grey and white matter of the FGR brain [21]. However, this current study does not examine inflammatory mediators; further investigation into the regulation of cytokine expression may reveal deeper insight into the underlying mechanisms that trigger neuroinflammation and contribute to the neuropathology of the PFC and BG. Furthermore, casual links to neuroinflammation were not directly tested in the current study; therefore, alternative mechanisms may also be at play, such as oxidative stress and excitotoxicity.

This study focused on post-mortem assessment of cellular pathology. Understanding the functional and behavioural outcomes associated with the described changes would add to our understanding of brain pathology in the newborn FGR, as described in smaller animal models [23]. This would be critical for translational implications as we may be able to discern how these cellular changes may relate to executive and motor dysfunction in infants with FGR. Our findings did not observe significant sex effects in the neuronal, white matter, and glial cells between male and female FGR piglets. However, there was an insufficient sample size to detect sex differences, and also an unequal representation of sexes. Previous studies have reported sex differences, with males displaying reductions only in MBP expressions compared to female FGR guinea pigs in the hippocampus at P1 [53]. Further studies with larger, balanced cohorts and specific sex-based analyses are required to elucidate potential sex-specific effects in the FGR pig model.

We have previously reported neuronal and white matter alterations in the pig parietal cortex during early postnatal life (P1–P4). The present study has extended the examination of neuropathology to a later time point and investigated regions not previously characterised. Our findings demonstrate less overt differences in both grey- and white-matter markers, with respect to relative differences in cell count and density when comparing FGR with NG brains. We note ongoing glial activation, which may be associated with sustained neuroinflammatory responses, but proinflammatory cytokines were not assessed in the present study. These findings indicate a degree of normalisation that may be associated with the observed reduction in cells undergoing apoptosis at this latter time point. However, at this stage, we cannot definitively determine if these changes are due to altered inflammatory profile, brain maturation, or region-specific vulnerability.

## 5. Conclusions

This study demonstrates that persistent grey and white matter impairments observed at postnatal day ten are associated with ongoing cell death and a sustained neuroinflammatory response, in accordance with neuropathological changes reported at earlier postnatal time points. However, as mentioned above, even though these changes are persistent at P10, they are not as pronounced as at P1 and P4. These cellular alterations were observed in the grey- and white-matter structures of the PFC and BG and suggest that the effect of FGR may be global, not region-specific, in the FGR brain. Ongoing glial activation and increased astrocytic death may play a key role in the progression of pathology. These observations indicate a link between glial activation and the persistent neuroinflammatory response. This highlights the opportunity for targeted therapeutic strategies to improve brain repair. Further studies focused on neuroinflammation could pave the way for developing targeted therapies to treat FGR-affected brains and improve long-term neurodevelopmental outcomes.

## Figures and Tables

**Figure 1 cells-14-01776-f001:**
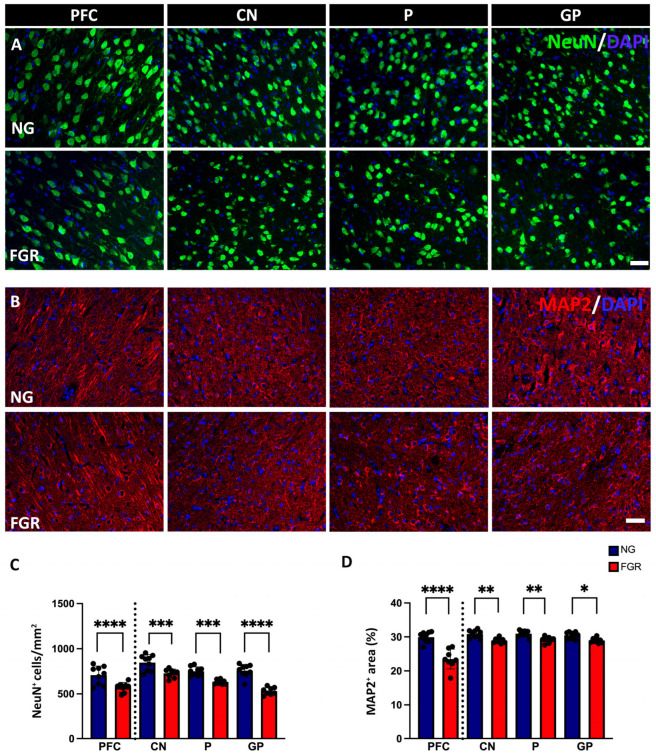
Neuronal and structural alterations associated with FGR brains at P10. (**A**) Representative labelling of neurons in the PFC and BG at P10. NeuN (green) shows intense labelling of mature neurons in NG brains. FGR group shows a region that is sparse of labelling in both PFC and BG. (**B**) Microtubule-associated protein 2 (MAP2: red) displays robust, well-structured cell bodies and dendrites. FGR group displays fragile perikaryal and dendritic MAP2 labelling compared with NG. (**C**) Quantification of neuronal cells demonstrates decreased cell numbers of NeuN-positive cells in FGR piglets compared with the NG brains. (**D**) MAP2-labelled area coverage is significantly decreased in FGR compared with NG brains in both the PFC and BG. Blue indicates NG, and orange indicates FGR piglets in the graphical representation. All values are expressed as the mean ± SEM (FGR = 8 piglets; NG = 9 piglets). Two-way ANOVA with Tukey’s post hoc test was used for this analysis (* *p* < 0.05, ** *p* < 0.01, *** *p* < 0.001, **** *p* < 0.0001) (scale bars: 50 μm).

**Figure 2 cells-14-01776-f002:**
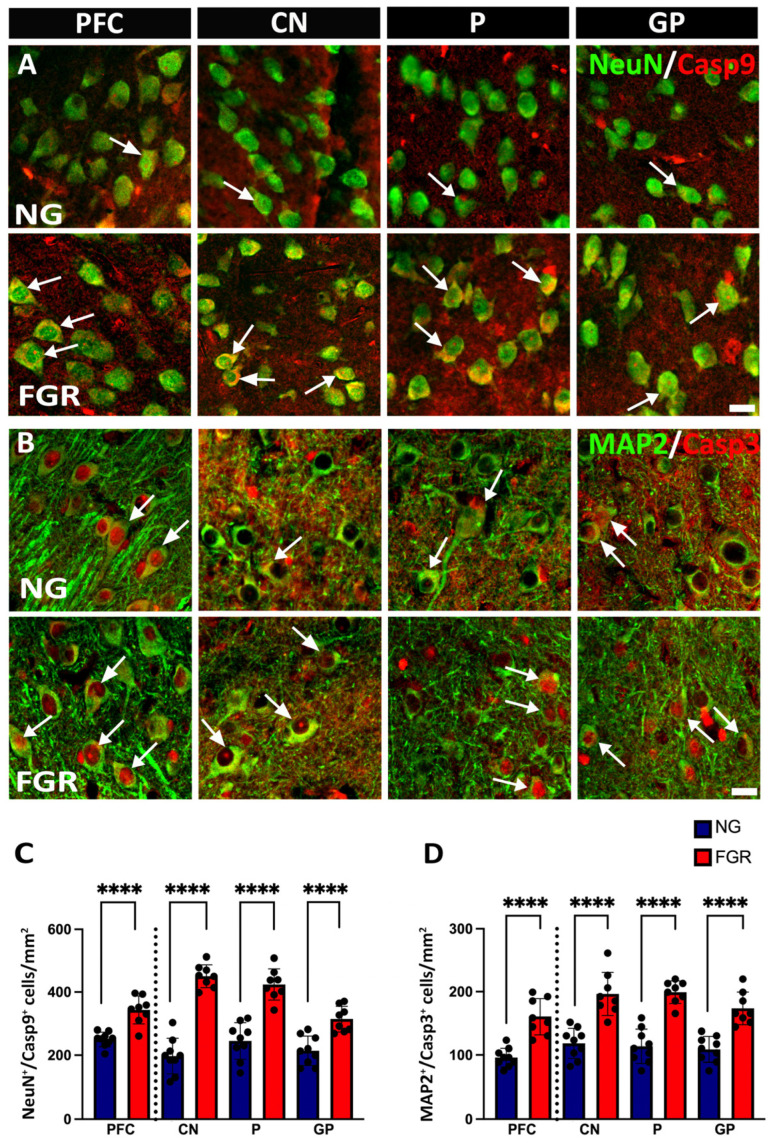
Upregulation of apoptotic activity in neuronal cells in the PFC and BG of FGR brains at P10. (**A**) Representative image of NG brains showing healthy mature neuronal cells (NeuN-positive cells: green) and clear co-localisation of Casp-9-positive (red) with mature neuronal cells. FGR brains show a greater amount of Casp-9-positive activities with the mature neuronal cells in the grey matter compared to NG (arrow indication). (**B**) NG brains display co-localisation of neuronal cells (MAP2-positive cells: green) with cleaved Casp-3-positive cells (red) in the grey matter of the PFC and BG. FGR brains show a high amount of co-localised MAP2-positive and Casp-3-positive cells (arrow indication). (**C**) Quantification of NeuN-positive/Casp-9-positive (NeuN^+^/Casp9^+^) cells demonstrates a significant increase in co-localisation in the grey matter of FGR brains. (**D**) Analysis of the co-localisation of MAP2-positive/Casp-3-positive (MAP2^+^/Casp3^+^) cells shows a significant increase in the grey matter of FGR brains. Blue indicates NG, and orange indicates FGR piglets in the graphical representation. All values are expressed as the mean ± SEM (FGR = 8 piglets; NG = 9 piglets). Two-way ANOVA with Tukey’s post hoc test was used for this analysis (**** *p* < 0.0001) (scale bars: 50 μm).

**Figure 3 cells-14-01776-f003:**
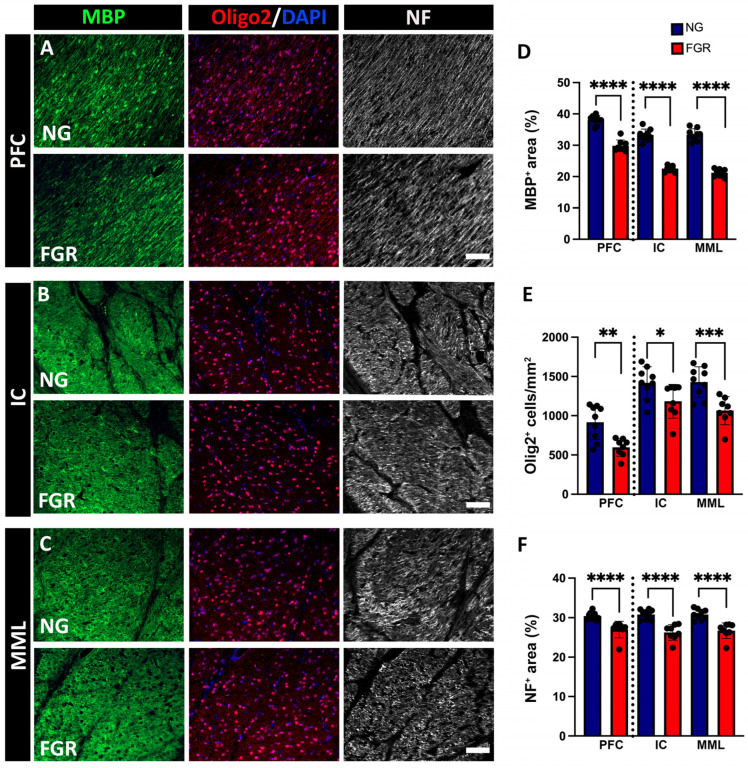
Impaired myelination in the white matter of FGR piglets at P10. (**A**) Representative image of NG piglets displaying robust and consistent labelling of myelin (MBP-positive), and neurofilament (NF-positive), with a high amount of oligodendrocytes (Olig2-positive) in the PFC. FGR piglets show interrupted labelling of MBP and NF with decreased Olig2-positive cells in the PFC. (**B**) NG group shows a high intensity of MBP and NF with increased amounts of Olig2-positive cells in the internal capsule region when compared to FGR. (**C**) Similar trends were observed, with NG group showing a higher intensity of MBP-positive and NF-positive cells, accompanied by an increase in the amount of Olig2-positive cells in the median medullary lamina. (**D**,**F**) Measurement of labelled area coverage (%) of MBP-positive and NF-positive, showing a significant decrease in the FGR white matter of both the PFC and BG. (**E**) FGR brain demonstrates a decreased number of Olig2-positive cells in both the PFC and BG compared to NG. Blue indicates NG, and orange indicates FGR piglets in the graphical representation. All values are expressed as the mean ± SEM (FGR = 8 piglets; NG = 9 piglets). Two-way ANOVA with Tukey’s post hoc test was used for this analysis (* *p* < 0.05, ** *p* < 0.01, *** *p* < 0.001, **** *p* < 0.0001) (scale bars: 50 μm).

**Figure 4 cells-14-01776-f004:**
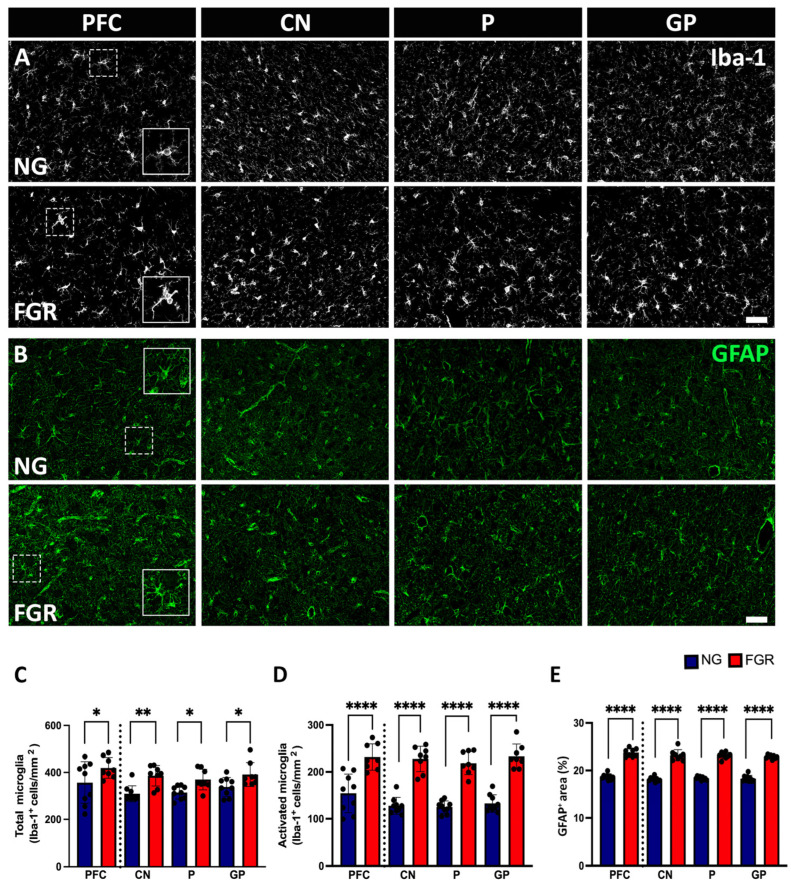
Glial cells in the grey matter of the PFC and BG at P10. (**A**) Representative labelling of NG brains shows a high amount of resting/ramified microglial cells (Iba-1: white) with small cell bodies possessing long, thin processes distributed throughout the grey matter of the PFC and BG (insert) as compared to FGR brains. FGR brains show a higher amount of ameboid-shaped activated microglial cells (insert) with retracted, dense cellular bodies in the grey matter of the PFC and BG compared with NG brains. (**B**) GFAP-labelled (green) astrocytes demonstrated more star-like cells with long process extensions (insert) in the grey matter of NG brains, whereas FGR brains show retracted GFAP-labelled cell bodies, thickened processes, indicating the reactive astrogliosis state (insert). (**C**) Quantification of the total number of microglial cells shows a significant increase in the grey matter of the BG. (**D**) Quantification of activated microglial cells revealed a significant increase in the grey matter of the FGR brains. (**E**) Quantification of astrocyte expression measured using densitometry (moments) reveals a significant increase in the GFAP-positive expression in the grey matter of FGR brains compared to NG. Blue indicates NG, and orange indicates FGR piglets in the graphical representation. All values are expressed as the mean ± SEM (FGR = 8 piglets; NG = 9 piglets). Two-way ANOVA with Tukey’s post hoc test was used for this analysis (* *p* < 0.05, ** *p* < 0.01, **** *p* < 0.0001) (scale bars: 50 μm).

**Figure 5 cells-14-01776-f005:**
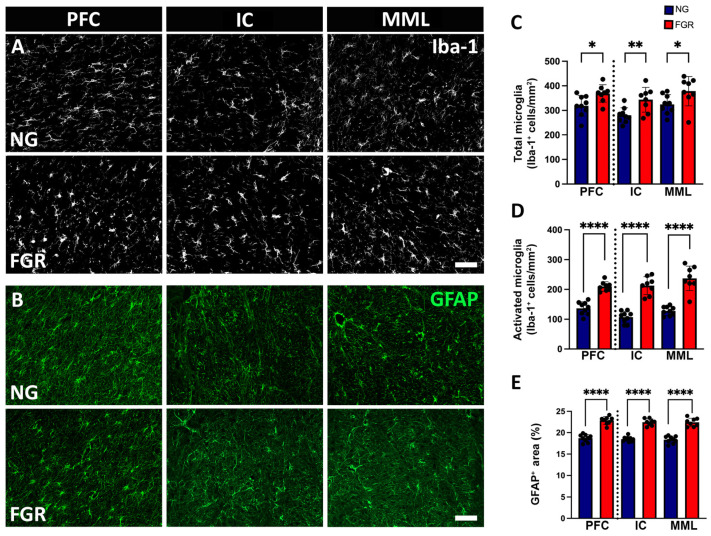
Glial cells in the white matter of the PFC and BG at P10. (**A**) Representative labelling of NG brains displays a higher amount of ramified/resting microglial cells throughout the white matter of the PFC and BG. FGR brains demonstrated an increased amount of activated/amoeboid-shaped cells in the white matter of the PFC and BG when compared to NG. (**B**) White matter of the PFC and the BG show a similar trend of GFAP-positive labelled area observed in the grey matter of the PFC and BG. (**C**) Quantification of the total number of microglial cells shows an increased number of microglial cells in the white matter of the PFC and BG. (**D**) Activated microglial cell count displays a significant increase in the white matter of the PFC and BG. (**E**) White matter of the PFC and BG shows a similar trend of significant increase in the GFAP-positive area coverage, as observed in the grey matter of the PFC and BG. Blue indicates NG, and orange indicates FGR piglets in the graphical representation. All values are expressed as the mean ± SEM (FGR = 8 piglets; NG = 9 piglets). Two-way ANOVA with Tukey’s post hoc test was used for this analysis (* *p* < 0.05 **, *p* < 0.01, **** *p* < 0.0001) (scale bars: 50 μm).

**Figure 6 cells-14-01776-f006:**
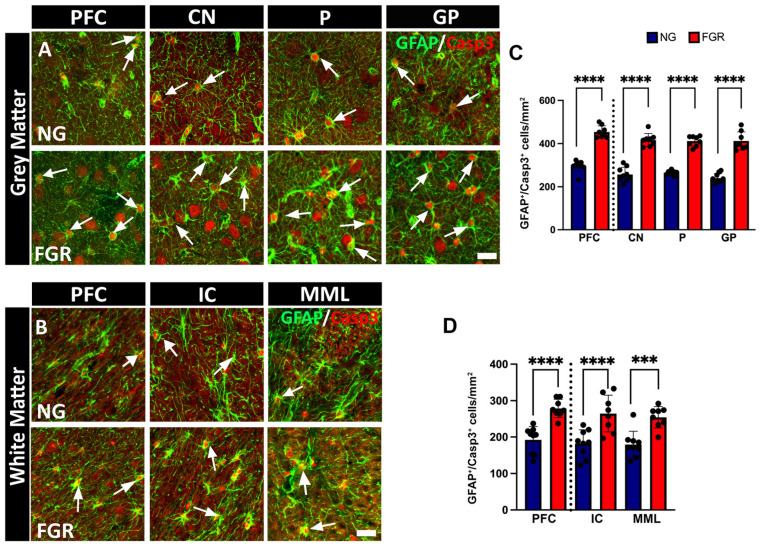
Increased apoptotic activity in astrocytic cells in both the grey and white matter of the PFC and BG of FGR brains at P10. (**A**) Representative image of the grey matter of NG brains displaying a low amount of co-localisation of GFAP-positive astrocytes (green) with Casp-3-positive cells (red) in both the PFC and BG (arrow indication) compared with FGR brains. (**B**) Similar findings were observed in the white matter, with a lower amount of GFAP-positive astrocytes co-localised with Casp-3-positive cells in both the PFC and BG of NG brains compared with FGR brains (arrow indication). (**C**,**D**) Quantification of co-localisation of GFAP-positive/Casp3-positive (GFAP^+^/Casp3^+^) cells shows a significant increase in the grey matter as well as the white matter of FGR brains. Blue indicates NG, and orange indicates FGR piglets in the graphical representation. All values are expressed as mean ± SEM (FGR = 8 piglets; NG = 9 piglets). Two-way ANOVA with Tukey’s post hoc test was used for this analysis (*** *p* < 0.001, **** *p* < 0.0001) (scale bars: 50 µm).

**Table 1 cells-14-01776-t001:** Piglet body weight and brain weight at postnatal day 10.

Piglets	NG Group (n = 9)	FGR Group (n = 8)
Body weight in kilograms (mean ± SEM)	2.594 ± 0.1927	1.406 ± 0.6555 ***
Brain weight in grams (mean ± SEM)	38.45 ± 1.143	31.83 ± 0.6285 ***
Brain-to-body weight (g/Kg)	15.44 ± 1.111	22.97 ± 1.069 ***

All values are expressed as mean ± SEM. Student’s *t*-test was used for this analysis (*** *p* < 0.001).

## Data Availability

The original contributions presented in this study are included in the article. Further inquiries can be directed to the corresponding author.

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
