# Peer review of "Foetal Growth Restriction Effects on Grey and White Matter in the Prefrontal Cortex and Basal Ganglia of Postnatal Day 10 Piglets"

_cells, 2025, doi:10.3390/cells14221776_

Round 1
Reviewer 1 Report
Comments and Suggestions for Authors
The submitted study is a morphofunctional study that evaluates cellular changes in the prefrontal cortex and basal ganglia in a piglet model of Fetal growth restriction (FGR) piglets (n=8 Fetal growth restriction affected vs n=9 controls sacrificed at Post partum day10). Glial cell activation, increased apoptosis, and alterations in neuronal and white matter have been reported in P10 FGR piglet brains.
The study uses immunofluorescence analysis (only) to evaluate in morphofuctional differences in different brain areas from FGR affected and controls; this may represent a limitation of the study , since the lack of any molecular analysis to further confirm morphofuctional data. Nevertheless, images have good quality, protocol is good and the provided data support the results.
My only query concerns the number of slices analysed for each brain area and information on the the sex of piglets.
Furthermore, this reviewer is not sure reference list has been formatted accordingly the journal style.
Reviewer 2 Report
Comments and Suggestions for Authors
The study claims novelty by focusing on the PFC and BG; however, similar apoptotic and glial activation patterns have already been reported in other brain regions in the same FGR piglet model (by the same group). The authors should clarify what unique insights this study provides about these specific regions (e.g., are PFC and BG more or less affected than previously studied regions?). It would be valuable to compare quantitatively how the P10 results differ from those earlier time points (P1 – P4) do they hypothesize that apoptosis decreases, persists, or increases with age? The translational implications (e.g., how these cellular changes might relate to executive and motor dysfunction in FGR infants) could be elaborated further in the Discussion. The study attributes changes to “persistent apoptosis and neuroinflammation,” but causal links are not directly tested. The authors should temper this interpretation or discuss potential alternative mechanisms.
The study includes 8 FGR and 9 NG piglets, which is small given inter-animal variability (especially in outbred animal models). Although typical for large-animal work, a power calculation or justification for sample size should be included to support the reliability of observed differences. The authors state that sex differences were not found, but the sample was not balanced by sex. Please specify the male/female ratio and discuss whether unequal representation could mask sex-specific effects. How are the authors certain that the FGR pups are not from the same maternal animal (in which case the SGA phenotype could be related to other etiologies (chorioamnionitis…) than the supposed placental insufficiency. Could the birth weight data on day 1 be provided?
The immunohistochemical quantifications rely on manual counting and ImageJ thresholding. The authors should specify how many fields per region per animal were analyzed, and whether analyses were performed bilaterally or unilaterally (I could read that all immunolabelling was performed in duplicate per animal, but no other details.
The use of multiple unpaired t-tests for each marker increases the risk of Type I error. A correction for multiple comparisons should be considered or at least discussed.
